# Serum procalcitonin as an independent diagnostic markers of bacteremia in febrile patients with hematologic malignancies

**Mina Yang**[1,2], **Seung Jun Choi**[3], **Jaewoong Lee**[1], **Dong Gun Lee**[4], **Yoon-Joo Kim**[5], **Yeon-Joon Park**[1], **Eun-Jee Oh**[1]*

1 Department of Laboratory Medicine, Seoul St. Mary's Hospital, College of Medicine, The Catholic University of Korea, Seoul, Korea, 2 Department of Laboratory Medicine, Samsung Changwon Hospital, Sungkyunkwan University School of Medicine, Changwon, Korea, 3 Department of Laboratory Medicine, Severance Hospital, Yonsei University College of Medicine, Seoul, Korea, 4 Department of Internal Medicine, Division of infection, Seoul St. Mary's Hospital, College of Medicine, The Catholic University of Korea, Seoul, Korea, 5 EONE Laboratories, Incheon, Korea

* ejoh@catholic.ac.kr

## Abstract

### Background

Serum procalcitonin (PCT) and C-reactive protein (CRP) are biomarkers of infection. In patients with hematologic disorders with or without hematopoietic stem cell transplantation (HSCT), it is difficult to distinguish bloodstream infections from aseptic causes of febrile episodes. The objective of this study was to investigate diagnostic values of PCT and CRP in predicting systemic bacterial infection in patients with hematologic malignancies.

### Methods

Clinical and laboratory data of 614 febrile episode cases from 511 patients were analyzed. Febrile episodes were classified into four groups: (1) culture-positive bacterial infection by Gram-positive cocci (GPC), (2) culture-positive bacterial infection by Gram-negative bacilli (GNB), (3) fungal infection, and (4) viral infection or a noninfectious etiology.

### Results

Of 614 febrile cases, systemic bacterial infections were confirmed in 99 (16.1%) febrile episodes, including 38 (6.2%) GPC and 61 (9.9%) GNB infections. PCT levels were significantly higher in GNB infectious episodes than those in febrile episodes caused by fungal infection (0.58 ng/mL (95% CI: 0.26–1.61) vs. 0.22 ng/mL (0.16–0.38), $P = 0.047$). Bacterial infectious episodes showed higher PCT and CRP levels than non-bacterial events (PCT: 0.49 (0.26–0.93) ng/mL vs. 0.20 (0.18–0.22) ng/mL, $P < 0.001$; CRP: 76.6 (50.5–92.8) mg/L vs. 58.0 (51.1–66.5) mg/L, $P = 0.036$). For non-neutropenic febrile episodes, both PCT and CRP discriminated bacteremia from non-bacteremia. However, in neutropenic febrile episodes, PCT only distinguished bacteremia from non-bacteremia. In non-neutropenic episode, both PCT and CRP showed good diagnostic accuracy (AUC: 0.757 vs. 0.763). In febrile neutropenia, only PCT discriminated bacteremia from non-bacterial infection (AUC:

**Data Availability Statement:** All relevant data are within the paper.

**Funding:** This work was supported by the National Research Foundation of Korea (NRF) grant funded

by the Korea government (MSIP) (NRF-2017R1A2B4011181), Republic of Korea. The funder had no role in study design, data collection and analysis, decision to publish, or preparation of the manuscript and only provided financial support in the form of research materials. EONE Laboratories provided support in the form of salaries for YJ.K., but did not have any additional role in the study design, data collection and analysis, decision to publish, or preparation of the manuscript. The specific roles of these authors are articulated in the 'author contributions' section.

**Competing interests:** This work was supported by the National Research Foundation of Korea (NRF) grant funded by the Korea government (MSIP) (NRF-2017R1A2B4011181), Republic of Korea. EONE Laboratories provided support in the form of salaries for YJ.K., but did not have any additional role in the study design, data collection and analysis, decision to publish, or preparation of the manuscript. This commercial affiliation had no role with any other relevant declarations relating to employment, consultancy, patents, products in development, or marketed products, etc. This does not alter out adherence to PLOS ONE policies on sharing data and materials. All authors declare all potential competing interests for the purposes of transparency.

0.624) whereas CRP could not detect bacteremia (AUC: 0.500, 95% CI: 0.439–0.561, $P > 0.05$).

## Conclusions

In this single-center observational study, PCT was more valuable than CRP for discriminating between bacteremia and non-bacteremia independent of neutropenia or HSCT.

## Introduction

Infectious complications remain a major issue in patients with hematological malignancy following chemotherapy or hematopoietic stem cell transplantation (HSCT). The key manifestation of infection is fever, although various noninfectious febrile episodes can also develop frequently. In HSCT patients, it is more complex to distinguish between infectious condition and aseptic causes of febrile events due to transplantation-related complications such as graft-versus-host disease, engraftment syndrome, thrombotic microangiopathy, and relapse of underlying diseases [1]. Early distinction of fever is needed to provide immediate antibiotic treatment. Therefore, in patients with suspicion of systemic bacterial infection, timely and adequate clinical decision making is important and blood culture is recommended [2–4].

C-reactive protein (CRP) and procalcitonin (PCT) are widely used biomarkers of infections. However, CRP levels are frequently increased in non-infectious complications. They show low specificity for infection, especially in patients with hematologic malignancies [5–8]. PCT is useful for the diagnosis of sepsis. In the presence of bacterial infection, PCT is rapidly produced by the C cells of the thyroid gland as well as several other cell type. PCT production is stimulated by two mechanisms, directly by bacterial endotoxins and lipopolysaccharides and indirectly by inflammatory mediators such as tumor necrosis factor-alpha, interleukin-6, interleukin-1 [9, 10]. It is known as a valuable biomarker for detecting bacterial infections with high specificity [1, 5, 6, 11]. Several studies have shown that PCT can discriminate etiologies of infection in patients with sepsis [12–14]. Koya et al. [1] have demonstrated that PCT could provide information for discriminating between bacterial or fungal infection and other causes. It could also predict patient's prognosis after HSCT [1, 10]. It has been also suggested that PCT could discriminate different etiologies of infection, namely Gram-positive cocci (GPC), Gram-negative bacilli (GNB), and fungus [15, 16]. However, whether PCT can discriminate bacterial infection from other etiologies of fever in patients with hematologic disorder remains controversial. In addition, studies about its usefulness and cut-off values for culture-positive bacteremia in large number of patients with hematologic malignancy are limited.

Thus, the objective of the present study was to retrospectively analyze 614 febrile episodes that developed in patients with hematologic malignancies and investigate diagnostic values of PCT and CRP in predicting systemic bacterial infection.

## Materials and methods

### Patients and clinical diagnosis

Patients with hematological malignancies and febrile episode who were admitted to Seoul St. May's hospital between February 2017 and June 2017 were considered for inclusion. We included 614 febrile episodes from 551 patients who had all laboratory data for PCT, CRP, and serial results of blood culture at the same time of febrile event. Fever was defined as an axillary

body temperature above 37.5˚C. Only initial febrile events after non-fever period of 1 week were included. Bacterial infection was defined as positive result of blood culture for bacteria except for coagulase-negative staphylococci. This study was approved by Institutional Review Board (approval number: KC18RESI0526) of Seoul. St. Mary's hospital, Seoul, Korea. Informed consent was waived by the board because the present retrospective study was performed using medical records.

Febrile episodes were classified into four groups according to culture results: (1) culture-positive bacterial infection by Gram-positive cocci (GPC), (2) culture-positive bacterial infection by Gram-negative bacilli (GNB), (3) culture-positive fungal infection or positive-aspergillus antigen assay with clinical symptoms (Fungus), and (4) viral infection or a noninfectious etiology including underlying disease, tumor lysis syndrome, drug, immune reaction or GVHD (Others). Group (1) and group (2) were classified as bacteremia (+) episodes while group (3) and group (4) were classified as bacteremia (-) episodes. Mixed infections [bacteremia (+) and culture-positive fungal infection (+)] were excluded from analysis.

## Laboratory tests

For each febrile episode, blood samples were collected within 24 hours after development of fever to measure serum PCT and CRP levels. Serum PCT levels were measured with fully automated chemiluminescent immunoassay using ADVIA Centaur B.R.A.H.M.S PCT (Siemens Healthcare Diagnostics, Berlin, Germany) according to the manufacturer's instructions. Serum CRP concentrations were measured using commercial turbidimetric immunoassay. Blood culture results were attained using a BACTEC FX automated blood culture system (Becton Dickinson, Sparks, MD, USA). When only one set of coagulase-negative staphylococci was detected, it was considered as contamination and a negative blood culture. Aspergillus antigen assay was performed using Platelia™ Aspergillus antigen immunoassay (Bio-Rad Laboratories, Marnes-la-Coquette, France).

## Statistical analysis

Results are described as median and range or 95% confidence interval (95% CI) for continuous variables. For categorical data, results are described as number and percentages. Comparisons were made using Chi-square test for categorical data and Mann-Whitney U test for non-normally distributed variables. Logistic regression analyses were used to determine independent variables predicting bacteremia in hematologic patients with febrile episode. Variables with $P$ value < 0.1 in univariate analysis were entered into logistic regression analysis using backward stepwise selection as described previously [17]. Diagnostic reliabilities of PCT and CRP for bacteremia were evaluated using receiver-operating characteristic (ROC) curve and area under the curve (AUC). Diagnostic accuracies including sensitivity and specificity were calculated using several cut-off levels. Optimal cutoff levels to detect bacteremia were determined using Youden's index. All analyses were conducted using SPSS software version 24.0 (IBM Corp., Armonk, NY, USA) and MedCalc version 19.0 (MedCalc, Mariakerke, Belgium). A $P$ value of less than 0.05 was considered statistically significant.

## Results

### Characteristics of febrile episodes

In patients with febrile episodes, leukemia (57.8%), including acute lymphoblastic leukemia (ALL), acute myeloid leukemia (AML), and chronic myeloid leukemia (CML), was the most frequent diagnosis followed by lymphoma (18.6%). Of 614 febrile episodes, 325 (52.9%)

episodes occurred in patients who underwent HSCT and 273 (44.5%) episodes happened in neutropenic period defined as absolute neutrophil count (ANC) $< 0.5 \times 10^9$/L. The primary diagnoses of 325 HSCT patients were AML (n = 135), ALL (n = 75), lymphoma (n = 44), multiple myeloma (n = 44) and myelodysplastic syndrome (n = 17). Systemic bacterial infections were confirmed in 99 (16.1%) febrile episodes, including 38 (6.2%) GPC and 61 (9.9%) GNB infections. Non-bacterial infections were identified in 515 (83.9%) febrile episodes caused by fungal infection (n = 29, 4.7%), viral infection (n = 28, 4.6%) or other noninfectious etiology (n = 458, 74.6%). The virus infections belong to the cytomegalovirus (n = 19), Human herpes virus (n = 5) and influenza virus (n = 4). Noninfectious febrile etiologies included pulmonary complication (n = 113), local infection (n = 35), engraft syndrome (n = 13), enterocolitis (n = 11), graft versus host disease (n = 8), relapse (n = 8) and miscellaneous or unknown source (n = 270).

## Characteristics of patients with bacteremia

Among a total 614 febrile episodes, 99 (16.1%) events were from patients with systemic bacterial infection. Clinical features between patients with bacteremia and those without bacteremia were compared. Results are shown in Table 1. Age or sex was not associated with bacteremia ($P > 0.05$). Patients with underlying diseases of ALL and AML showed higher frequencies of bacteremia ($P = 0.033$ and $P = 0.009$, respectively). Of 99 febrile episodes with bacteremia, 65 episodes were from HSCT patients (20.0%, 65/325) and 34 episodes were from non-HSCT patients (11.8%, 34/289) ($P = 0.006$). In univariate logistic regression analysis, neutropenia, HSCT, PCT level, and CRP level were significantly associated with bacteremia (all $P < 0.05$). Multivariate logistic regression analysis demonstrated that neutropenia and PCT level, but not CRP level, were significantly associated with bacteremia [odd ratio (95% CI): 8.220 (4.645–14.549) for neutropenia ($P < 0.001$) and 1.048 (1.025–1.071) for PCT ($P < 0.001$)].

**Table 1. Comparison of clinical features between bacteremia (+) and bacteremia (-) subgroups.**

| Characteristics | Bacteremia (-) (n = 515) | Bacteremia (+) (n = 99) | P-value |
|---|---|---|---|
| Sex (Male), no (%) | 278 (54.0) | 49 (49.5) | NS |
| Age, median (range) | 54 (20–90) | 54 (20–74) | NS |
| Underlying disease, no (%) | | | |
| ALL | 84 (16.3) | 25 (25.3) | 0.033 |
| AML | 169 (32.8) | 46 (46.5) | 0.009 |
| CML | 24 (4.7) | 7 (7.1) | NS |
| Lymphoma | 102 (19.8) | 12 (12.1) | 0.072 |
| MDS | 41 (8.0) | 1 (1.0) | 0.012 |
| MM | 73 (14.2) | 6 (6.1) | 0.027 |
| Others | 22 (4.3) | 2 (2.0) | NS |
| HSCT | 260 (50.5) | 65 (65.7) | 0.006 |
| Allo-HSCT | 180 (69.2) | 57 (87.7) | 0.003 |
| Auto-HSCT | 80 (30.8) | 8 (12.3) | |
| ANC (x10$^9$/L), mean/median (range) | 3.88/1.52 (0–194.3) | 0.81/0.00 (0–11.3) | <0.001 |
| Neutropenia (+), no (%) | 192 (37.3) | 81 (81.8) | <0.001 |
| PCT (ng/mL), median (95% CI) | 0.20 (0.18–0.22) | 0.49 (0.26–0.93) | <0.001 |
| CRP (mg/L), median (95% CI) | 58.0 (51.1–66.5) | 76.6 (50.5–92.8) | 0.036 |

ALL, acute lymphoblastic leukemia; AML, acute myeloid leukemia; ANC, absolute neutrophil count; CI, confidence interval; CML, chronic myeloid leukemia; CRP, C-reactive protein; HSCT, hematopoietic stem cell transplantation; MDS, myelodysplastic syndrome; MM, multiple myeloma; PCT, procalcitonin.

## Results of PCT and CRP levels to detect bacteremia in neutropenic and non-neutropenic patients

Ninety-nine systemic bacterial infection episodes showed higher PCT and CRP levels than nonbacterial events (PCT: 0.49 (0.26–0.93) ng/mL vs. 0.20 (0.18–0.22) ng/mL, $P < 0.001$; CRP: 76.6 (50.5–92.8) mg/L vs. 58.0 (51.1–66.5) mg/L, $P = 0.036$) (Fig 1). PCT and CRP levels were further analyzed between the presence and absence of neutropenia. In 341 non-neutropenic febrile episodes, both PCT and CRP levels discriminated bacteremia from non-bacteremia (PCT: 0.97 (0.27–5.61) ng/mL vs. 0.20 (0.17–0.24) ng/mL, $P < 0.001$; CRP: 146.1 (78.9–240.2) mg/L vs. 52.2 (46.4–62.0) mg/L, $P < 0.001$). In 273 neutropenic febrile episodes, PCT levels were higher in bacteremia compared to those in non-bacteremia [0.44 (0.19–0.78) ng/mL vs. 0.20 (0.18–0.22) ng/mL, P = 0.001] whereas CRP levels were not significantly different between bacteremia and non-bacteremia [59.7 (41.7–84.5) mg/L vs. 69.5 (54.6–79.0) mg/L, $P = 0.994$] (Fig 2).

## Results of PCT and CRP levels for bacteremia in HSCT and non-HSCT patients

PCT and CRP levels were separately evaluated in HSCT patients and non-HSCT patients. In 289 febrile episodes from non-HSCT patients, PCT and CRP levels were higher in 34 bacteremia cases than those in 255 non-bacteremia cases (PCT: 0.91 (0.27–5.51) ng/mL vs. 0.22 (0.18–0.28) ng/mL, $P < 0.001$; CRP: 125.7 (91.3–159.8) mg/L vs. 67.4 (55.7–81.9) mg/L, $P = 0.002$). In 325 febrile episodes form HSCT patients, PCT levels were higher in bacteremia cases compared to those in non-bacteremia cases [0.30 (0.18–0.58) ng/mL vs. 0.19 (0.16–0.22) ng/mL, $P < 0.001$] whereas CRP levels were not significantly different between bacteremia and non-bacteremia cases [51.0 (39.5–78.4) mg/L vs. 49.3 (39.9–61.3) mg/L, $P = 0.414$] (Fig 3). Neutropenia was more frequently found in HSCT patients than that in non-HSCT patients (186/325 (57.2%) vs. 87/289 (30.1%), $P < 0.001$). However, HSCT was not an independent factor for bacteremia in multivariate logistic regression analysis.

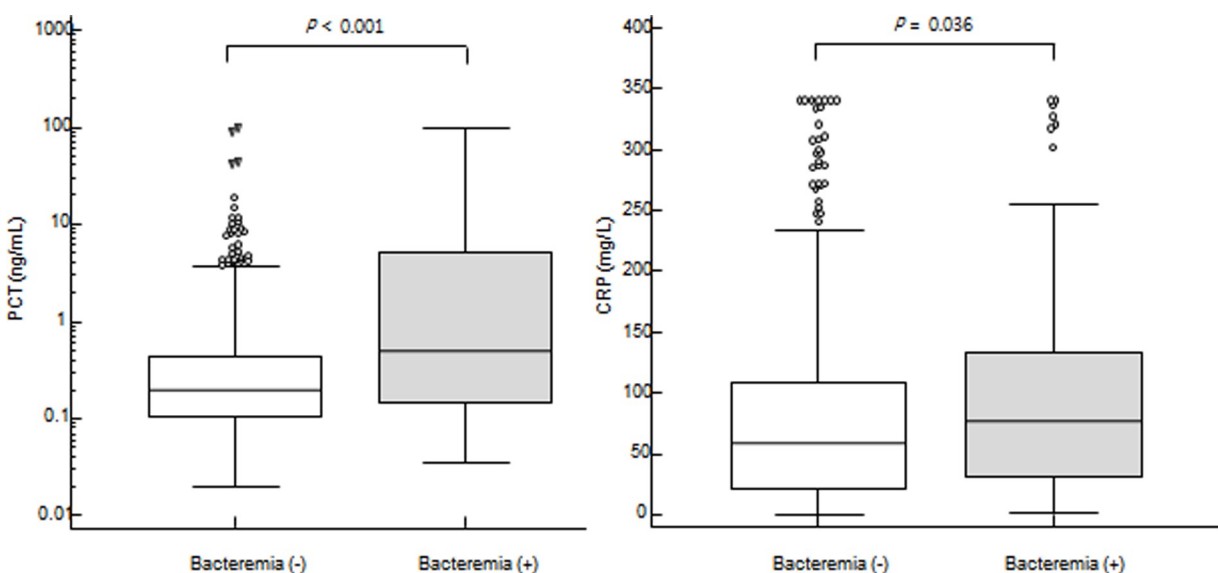

**Fig 1. Comparison of PCT (left) and CRP (right) levels between bacteremia and non-bacteremia.** Ninety-nine systemic bacterial infection episodes (colored box) showed higher PCT and CRP levels than nonbacterial events [median (95% CI) (PCT: 0.49 (0.26–0.93) ng/mL vs. 0.20 (0.18–0.22) ng/mL, $P < 0.001$; CRP: 76.6 (50.5–92.8) mg/L vs. 58.0 (51.1–66.5) mg/L, $P = 0.036$)] by Mann-Whitney U test.

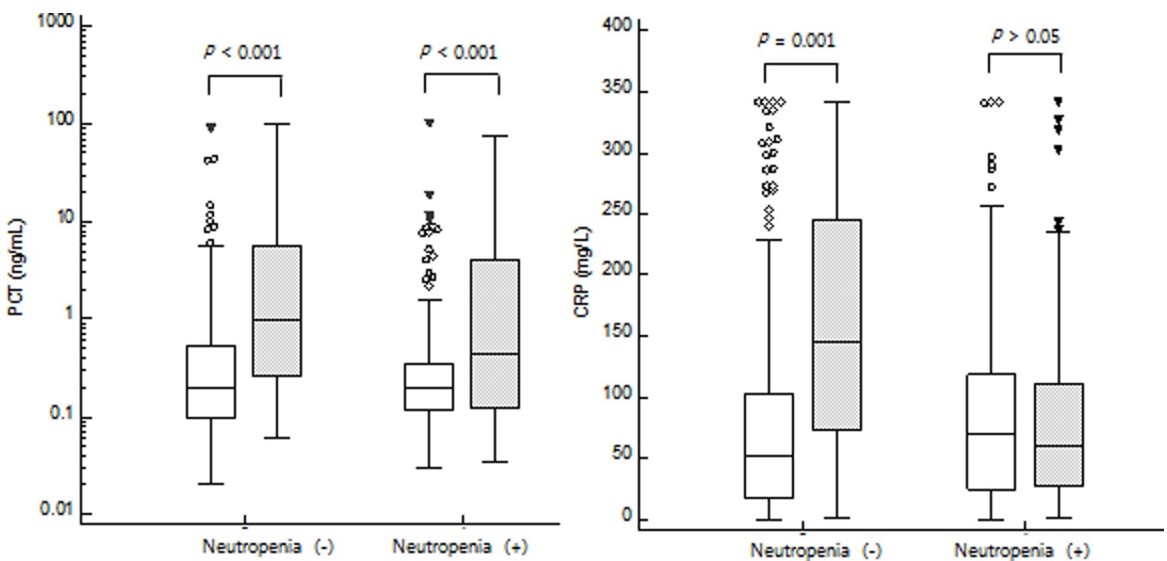

**Fig 2. Comparison of PCT (left) and CRP (right) levels between neutropenic and non-neutropenic patients.** PCT levels discriminated bacteremia (gray box) from non-bacteremia (white box) in both neutropenia (-) patients [median (95% CI), 0.97 (0.27–5.61) ng/mL vs. 0.20 (0.17–0.24) ng/mL, $P < 0.001$] and neutropenia (+) patients [0.44 (0.19–0.78) ng/mL vs. 0.20 (0.18–0.22) ng/mL, $P = 0.001$]. CRP levels discriminated bacteremia (gray box) from non-bacteremia (white box) in neutropenia (-) patients [146.1 (78.9–240.2) mg/L vs. 52.2 (46.4–62.0) mg/L, $P < 0.001$)], but CRP levels were not different between bacteremia and non-bacteremia in neutropenia (+) patients [59.7 (41.7–84.5) mg/L vs. 69.5 (54.6–79.0) mg/L, $P = 0.994$] by Mann-Whitney U test.

## Diagnostic accuracy of PCT and CRP for detecting bacteremia

ROC curves were generated for CRP and PCT levels to detect bacteremia in a total of 614 febrile episodes (Fig 4A), in non-neutropenic febrile episodes (Fig 4B), and in neutropenic

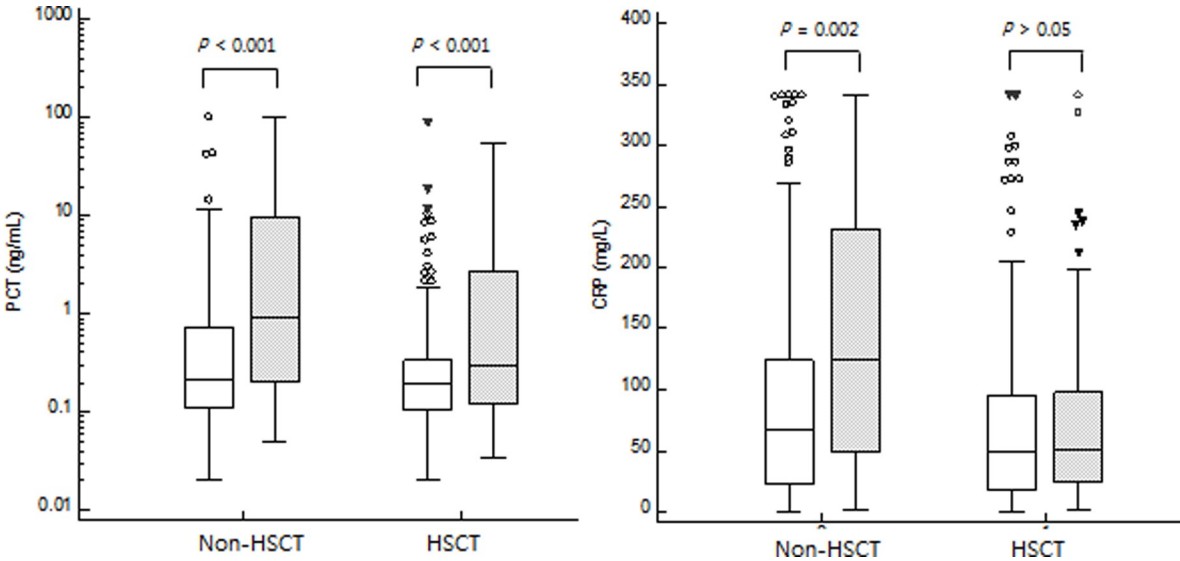

**Fig 3. Comparison of PCT (left) and CRP (right) levels between HSCT and non-HSCT.** PCT levels discriminated bacteremia (gray box) from non-bacteremia (white box) in both non-HSCT patient [median (95% CI), 0.91 (0.27–5.51) ng/mL vs. 0.22 (0.18–0.28) ng/mL, $P < 0.001$] and HSCT patients [0.30 (0.18–0.58) ng/mL vs. 0.19 (0.16–0.22) ng/mL, $P < 0.001$]. CRP levels discriminated bacteremia (gray box) from non-bacteremia (white box) in non-HSCT patient [125.7 (91.3–159.8) mg/L vs. 67.4 (55.7–81.9) mg/L, $P = 0.002$] but CRP levels were not different between bacteremia and non-bacteremia in HSCT patients [51.0 (39.5–78.4) mg/L vs. 49.3 (39.9–61.3) mg/L, $P = 0.414$] by Mann-Whitney U test.

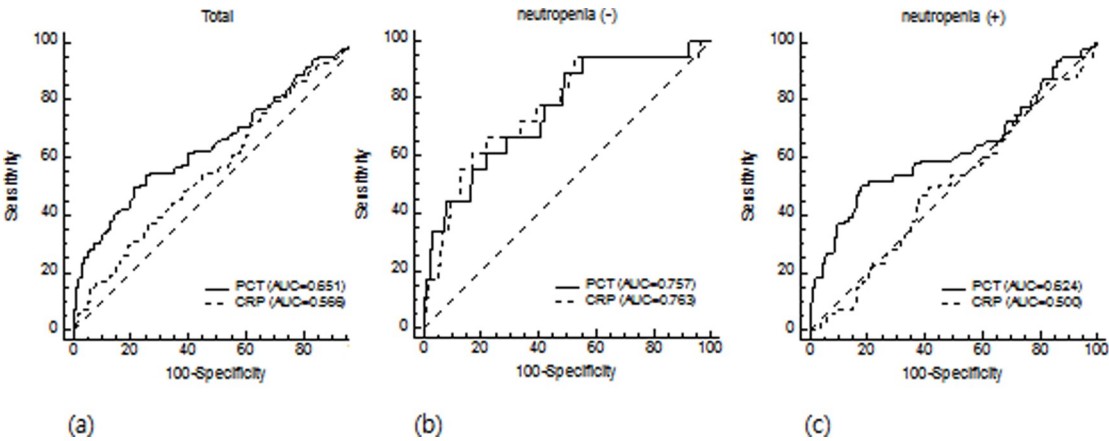

**Fig 4. ROC curves of PCT and CRP for detecting bacteremia in total (a), non-neutropenic (b), and neutropenic (c) patients.**
In a total of 614 febrile episodes (a), the AUC of PCT was 0.651 (95% CI: 0.612–0.689) while that of CRP was 0.566 (95% CI: 0.526–0.606). In 341 non-neutropenic episodes (b), PCT and CRP showed AUC of 0.757 (95% CI: 0.708–0.801) and 0.763 (95% CI: 0.714–0.807), respectively. In 273 febrile neutropenia (c), PCT discriminated bacteremia from non-bacterial infection (AUC: 0.624, 95% CI: 0.564–0.682) whereas CRP could not detect bacteremia (AUC: 0.500, 95% CI: 0.439–0.561, $P > 0.05$).

febrile episodes (Fig 4C). In a total of 614 febrile episodes, the AUC of PCT was 0.651 (95% CI: 0.612–0.689) while that of CRP was 0.566 (95% CI: 0.526–0.606), with PCT showing higher diagnostic performance than CRP in pairwise comparison ($P = 0.017$). With a cut-off of 0.5 ng/mL, PCT showed sensitivity of 49.5% and specificity of 79.0%. CRP with a cut-off of 25 mg/L showed sensitivity of 79.8% and specificity of 29.3% (Table 2). When we analyzed PCT and CRP levels in 341 non-neutropenic episodes, both PCT and CRP showed good diagnostic accuracies, with AUC of 0.757 (95% CI: 0.708–0.801) (88.9% sensitivity and 51.1% specificity using cut-off of 0.2 ng/mL) and 0.763 (95% CI: 0.714–0.807) (66.7% sensitivity and 73.7% specificity using cut-off of 100 mg/L), respectively. However, in 273 febrile neutropenia, only PCT discriminated bacteremia from non-bacterial infection (AUC: 0.624, 95% CI: 0.564–0.682) whereas CRP could not detect bacteremia (AUC: 0.500, 95% CI: 0.439–0.561, $P > 0.05$). Table 2 shows sensitivity and specificity of CRP and PCT for detecting bacteremia at different cut-off values.

## PCT and CRP levels according to different etiologies of fever

PCT and CRP levels in four different febrile etiologies (38 GPC, 61 GNB, 29 fungus and 486 others etiologies) are shown in Fig 5. PCT levels were significantly higher in GNB infectious episodes than those in febrile episodes caused by fungal infection [0.58 (95% CI: 0.26–1.61) vs. 0.22 (0.16–0.38), $P = 0.047$] or other etiology [0.58 (95% CI: 0.26–1.61) vs. 0.19 (0.17–0.22), $P < 0.001$]. Although bacteremia (+) patients had higher CRP levels than bacteremia (-) patients, CRP levels were not significantly different among the four groups ($P > 0.05$). There were no significant differences in PCT or CRP level between Gram-positive and Gram-negative infectious episodes (P > 0.05). Among identified GNBs including *Escherichia coli* (*E. coli*), Klebsiella species, and *Pseudomonas aeruginosa*, *E. coli* was the most frequently detected GNB. There was no significant difference in PCT level among causative GNB species.

## Discussion

In hematological patients, early differentiation of fever would be of significance. This retrospective study was implemented to evaluate performances of PCT and CRP to detect

**Table 2. Diagnostic values of PCT and CRP for bacteremia.**

| | PCT | | | CRP | | |
|---|---|---|---|---|---|---|
| | Cut-off (ng/mL) | Sensitivity (%) | Specificity (%) | Cut-off (mg/L) | Sensitivity (%) | Specificity (%) |
| All febrile episodes (n = 614) | 0.1 | 86.9 | 22.9 | 25* | 79.8 | 29.3 |
| | 0.2 | 64.7 | 50.9 | 50 | 60.6 | 44.9 |
| | 0.5* | 49.5 | 79.0 | 100 | 37.4 | 72.4 |
| | 1.0 | 38.4 | 86.0 | 150 | 19.2 | 85.8 |
| Non-neutropenic episodes (n = 341) | 0.1 | 94.4 | 25.1 | 25 | 94.4 | 31.9 |
| | 0.2* | 88.9 | 51.1 | 50 | 94.4 | 47.1 |
| | 0.5 | 61.1 | 74.6 | 100* | 66.7 | 73.7 |
| | 1.0 | 44.4 | 83.9 | 150 | 44.4 | 87.3 |
| Febrile neutropenia (n = 273) | 0.1 | 85.2 | 19.3 | 25 | NA | NA |
| | 0.2 | 59.3 | 50.5 | 50 | NA | NA |
| | 0.5* | 46.9 | 83.3 | 100 | NA | NA |
| | 1.0 | 37.0 | 90.1 | 150 | NA | NA |

*optimal cut-off levels by Youden's index.

CRP, C-reactive protein; PCT, procalcitonin

bacteremia in 614 febrile episodes from patients with hematological malignancy. In the present study, bacteremia was defined as culture-positive bacterial infection in serial results of blood culture at febrile event, and finally, PCT had better diagnostic accuracy for bacteremia than CRP.

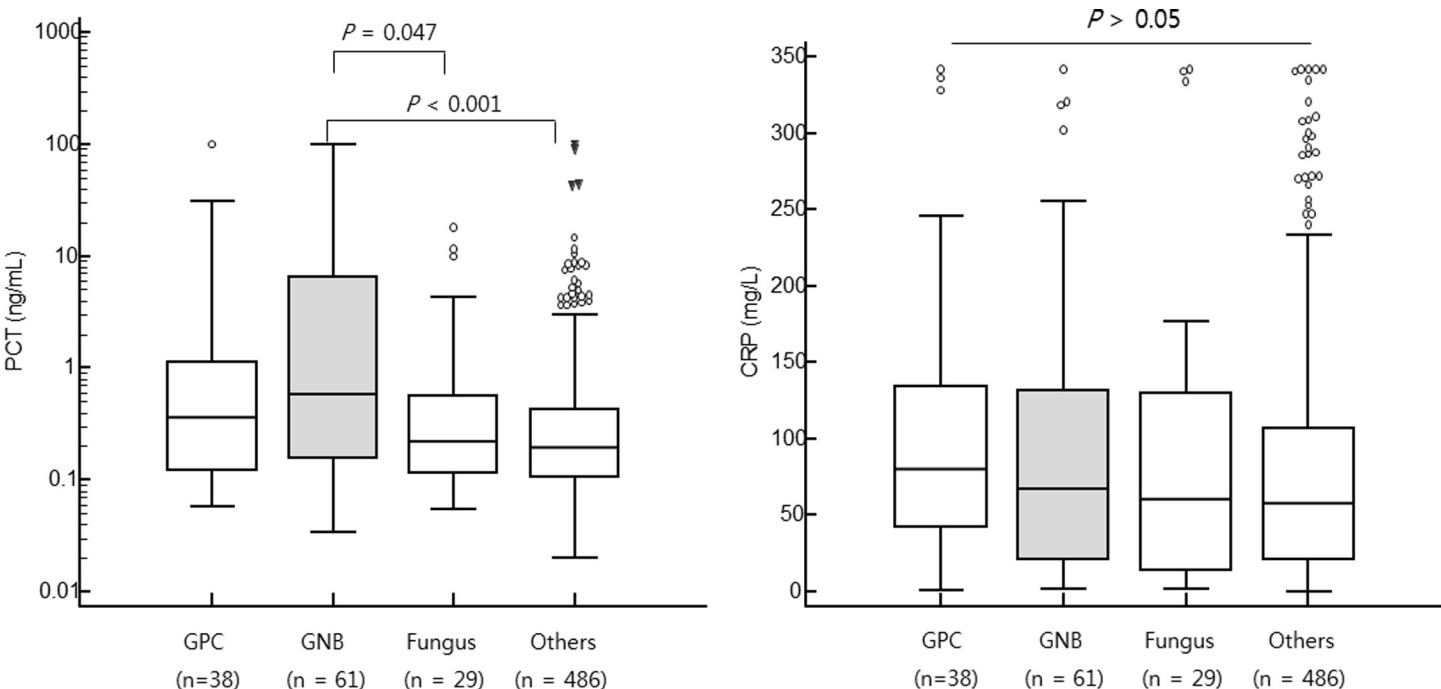

**Fig 5. Comparison of PCT (left) and CRP (right) levels among different etiologies of fever.** PCT levels were significantly higher in GNB infectious episodes (gray box) than those in febrile episodes caused by fungal infection [median (95% CI), 0.58 (95% CI: 0.26–1.61) vs. 0.22 (0.16–0.38), P = 0.047] or other etiology [0.58 (95% CI: 0.26–1.61) vs. 0.19 (0.17–0.22), P < 0.001]. CRP levels were not significantly different among the four groups (P > 0.05).

As a biomarker of inflammation, CRP has been widely used in clinical practice [6, 12, 18, 19]. However, it cannot adequately differentiate etiologies of fever in hematologic patients due to its low specificity [20]. Previous reports have shown that PCT has high specificity for bacterial infections, and PCT has been introduced into clinical use [21, 22]. Kim et al., reported that PCT showed better diagnostic value than CRP in febrile neutropenic patients with solid cancer [19]. Our study also confirmed that PCT had higher specificity than CRP in hematologic patients. This is consistent with a previous finding showing that PCT measurement has better diagnostic value than CRP in adult patients regardless of type of comorbidities [21].

Diagnostic utility of PCT in neutropenic patients remains controversial. Leukocytes are known to produce PCT and inflammatory cytokines released from leukocytes can mediate PCT production. Previous study has noted that PCT values greater than 0.5 ng/ml are less common in patients with neutropenia [23]. When we analyzed diagnostic accuracies of PCT and CRP in 273 febrile neutropenia, only PCT had diagnostic value (AUC of 0.624) with specificity of 83.3% using cut-off of 0.5 ng/mL. However, the sensitivity of PCT was relatively low (46.9%), consistent with results of a previous study demonstrating low sensitivity of PCT in febrile neutropenia [6]. Our data support previous results of Lima et al. [24] showing that PCT cut-off point of 0.5 ng/mL is correlated with bacteremia (sensitivity of 51.9% and specificity of 76.5%) in a randomized controlled trial enrolling 62 hematological adult patients who have febrile neutropenia. To obtain high sensitivity of PCT test, lower cut-off levels should be considered. Therefore, clinicians need to expect low PCT levels in patients with febrile neutropenia and utilize PCT to help them confirm bacteremia.

In HSCT patients, diagnostic values of PCT and CRP for detecting infections are controversial, and previous studies have focused on different target populations [1, 6, 8, 25]. A meta-analysis by Lyu et al. [26] has concluded that PCT has only a moderate diagnostic value in discriminating infection from other inflammatory complications following allogeneic HSCT. Mori et al. [6] have shown that CRP is a better indicator for infections than PCT in HSCT while PCT is a better diagnostic marker for infections than CRP in non-HSCT. Several studies have also suggested that HSCT-related complications including GVHD and immune reaction can induce PCT positivity by stimulating mononuclear cells to produce inflammatory cytokines such as TNF-α [27]. In the present study, only PCT showed diagnostic value for detecting bacteremia in HSCT patients. However, HSCT patients were associated with neutropenia while HSCT was not an independent factor for bacteremia in multivariate analysis.

In this study, we also aimed to evaluate whether serum PCT or CRP levels could differentiate infection etiologies in febrile patients with hematologic disorders. PCT levels, but not CRP levels, were significantly higher in patients with Gram-negative bacteremia than in patients with fungus or other non-bacterial etiology ($P < 0.05$). Other non-bacterial etiologies included viral infection and immunosuppressant or underlying disease-related febrile episodes. In viral infections, high concentration of interferon-gamma can suppress PCT production. Consequently, high level of PCT can be used for supportive diagnosis of bacterial infection [28].

In the present study, there was no significant difference in PCT level between Gram-negative and Gram-positive bacteremia, consistent with results of previous studies [6, 29]. However, this finding is not consistent with results of previous studies reporting that greater increase in PCT level was observed in *Enterobacteriaceae* compared to that in nonfermentative GNB. Leli et al. [15] have proposed that infection caused by *Enterobacteriaceae* is not likely to be associated with PCT level ≤ 3.1 ng/mL. They showed significant difference in PCT value in association with infection by nonfermentative GNB. Furthermore, Yan et al. [16] have reported that PCT could distinguish between different bacterial species and infection sites. In the present study, 31 (66.0%) of 47 bacteremia cases caused by *Enterobacteriaceae* showed PCT level ≤ 3.1 ng/mL and 20 episodes even showed PCT level < 0.5 ng/mL. Although

nonfermentative GNB cases had lower median PCT levels than cases caused by *Enterobacteria-ceae*, PCT levels were not significantly different between *Enterobacteriaceae* and nonfermentative GNB cases. The relatively low median PCT level for *Enterobacteriaceae* in this study could be explained by characteristics of neutropenic patients. While Leli et al. [15] and Yan et al. [16] included septic patients with heterogeneous underlying diseases, our study was undertaken in patients with hematologic malignancy who tended to have neutropenia due to chemotherapy or HSCT.

This study has some limitations. In this retrospective study, we could not demonstrate PCT or CRP levels in association with clinical course, specific transplant-related complications, or response to antimicrobial therapy. As previous studies reported that PCT was useful in monitoring the response to the infection and to the treatment [1, 10], prognostic role of PCT needs to be further explored in a large, well-defined cohort study. Despite these limitations, the major strength of this study was that it demonstrated the diagnostic value of PCT with different cut-off levels for culture-based bacteremia in large number of patients including patients with febrile neutropenia. This study on the diagnostic accuracy of PCT and CRP for bacteremia in hematologic patients including patients with neutropenia or HSCT is one of the largest studies conducted up to date.

In summary, for febrile episodes from patients with hematologic malignancy, PCT was more valuable than CRP for discriminating between bacteremia and non-bacteremia independent of neutropenia. PCT in combination with clinical parameters should be considered to predict bacteremia in these patients. Further studies are needed to validate the cut-off level of PCT in patients with different conditions.

## Author Contributions

**Conceptualization:** Eun-Jee Oh.

**Data curation:** Jaewoong Lee.

**Formal analysis:** Mina Yang, Seung Jun Choi, Jaewoong Lee.

**Investigation:** Eun-Jee Oh.

**Methodology:** Seung Jun Choi, Jaewoong Lee.

**Supervision:** Dong Gun Lee, Yeon-Joon Park, Eun-Jee Oh.

**Validation:** Yoon-Joo Kim.

**Writing – original draft:** Mina Yang.

**Writing – review & editing:** Eun-Jee Oh.

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
