## [Decision Letter · Decision Letter 0]

18 Sep 2019

PONE-D-19-16377

Serum procalcitonin as an independent diagnostic markers of bacteremia in febrile patients with hematologic malignancies

PLOS ONE

Dear Dr. Oh,

Thank you for submitting your manuscript to PLOS ONE. After careful consideration, we feel that it has merit but does not fully meet PLOS ONE’s publication criteria as it currently stands. Therefore, we invite you to submit a revised version of the manuscript that addresses the points raised during the review process.

We have now received reports from two referees of your manuscript, as agree with reviewers comments raised a few concerns about this study. After careful consideration, we invite you to submit a revised version of the manuscript

We would appreciate receiving your revised manuscript by Nov 02 2019 11:59PM. To enhance the reproducibility of your results, we recommend that if applicable you deposit your laboratory protocols in protocols.io, where a protocol can be assigned its own identifier (DOI) such that it can be cited independently in the future. For instructions see: http://journals.plos.org/plosone/s/submission-guidelines#loc-laboratory-protocols

We look forward to receiving your revised manuscript.

Kind regards,

Senthilnathan Palaniyandi, Ph.D

Academic Editor

PLOS ONE

Journal Requirements:

This work was supported by the National Research Foundation of Korea (NRF) grant funded by the Korea government (MSIP) (NRF-2017R1A2B4011181), Republic of Korea.The funder had no role in study design, data collection and analysis, decision to publish, or preparation of the manuscript.

We note that one or more of the authors are employed by a commercial company: EONE Laboratories.

Reviewers' comments:

Reviewer's Responses to Questions

**Comments to the Author**

1. Is the manuscript technically sound, and do the data support the conclusions?

Reviewer #1: Partly

Reviewer #2: Partly

2. Has the statistical analysis been performed appropriately and rigorously? 

Reviewer #1: Yes

Reviewer #2: Yes

3. Have the authors made all data underlying the findings in their manuscript fully available?

Reviewer #1: Yes

Reviewer #2: Yes

4. Is the manuscript presented in an intelligible fashion and written in standard English?

Reviewer #1: Yes

Reviewer #2: No

5. Review Comments to the Author

Reviewer #1: The manuscript of Dr. Eun-Jee Oh. and its group by `` Serum procalcitonin as an independent diagnostic markers 1 of bacteremia in febril patients with hematologic malignancies` entitle was studied.

- The subject and experiment design was good.

Some question in manuscript should be answered:

1- What is the mechanism of infection or fever to induce increase of PCT level?

2- Fever is responsible to increase serum level of PCT or PCT is responsible for fever?

3- However, usually patients with malignant hematopoiesis disorder have fever and immune system of this kind of patient is not effective, therefore infection is as usual. May be the cause of increase serum level of PCT, were not be related to malignant hematopoiesis disorder.

The Authors find that the PCT serum level increase by malignant hematopoiesis disorder or transplantation, here we should consider to some question:

a- In other malignancies what is happened for PCT? It has been necessary to have data from other malignancy and discusses about that.

b- Is it possible PCT would be prognosis factor or treatment evaluated factor?

The discussion should be more and improve.

Reviewer #2: The authors present an analysis of a large patient cohort with haematological diseases focusing on infection monitoring with laboratory test such as CRP and PCT in addition to microbiological examinations. However, the presentation of the data in its current form is poor, lacks clarity, important details and interpretation.

Major:

1. It should be elaborated if there are overlaps between patients with hematology diagnosis and those that were treated by HSCT. What were the typical diagnoses for HSCT?

2. More details should be provided about viral infections and non-infectious etiology: lines 145 and 146.

3. Statements of lines 163 and 164 do not correspond with the numbers of Table 2.

4. Figure legends in general must be substantially extended with additional details e.g. case numbers in the respective subgroups, rational and method of subgrouping, statistics employed etc. I do not clearly understand the difference between Figs 1 and 2, similarly the meaning of colour is also missing.

5. In addition to the above deficiencies, in the context of Fig 5 (lines 239 through 248), respective case numbers are missing. There is an apparent discrepancy between CRP differences in earlier comparisons and the apparent similarities in these comparisons.

6. The abstract should be completely rewritten. For me the line of study is unclear. In my opinion a single observation is arbitrarily emphasized as conclusion.

Minor: potential points of grammatical improvements

1. Table 1 should be omitted, it is redundant relative to Table 2

2. In Table 2 ANC value (median) <0.01 is unlikely to be correct for Bacteriemia pos pts

3. Table 2 abbreviations should be in alphabetical orde

4. English proficiency should be further improved. Even myself, a non-English native, found several incorrectly worded phrases.

6. PLOS authors have the option to publish the peer review history of their article (what does this mean?). If published, this will include your full peer review and any attached files.

Reviewer #1: No

Reviewer #2: No

---

## [Author Response · Author response to Decision Letter 0]

15 Oct 2019

Response to Reviewers

PONE-D-19-16377

Serum procalcitonin as an independent diagnostic markers of bacteremia in febrile patients with hematologic malignancies

PLOS ONE

Reviewer #1: The manuscript of Dr. Eun-Jee Oh. and its group by `` Serum procalcitonin as an independent diagnostic markers 1 of bacteremia in febril patients with hematologic malignancies` entitle was studied.

- The subject and experiment design was good.

Some question in manuscript should be answered:

1- What is the mechanism of infection or fever to induce increase of PCT level?

Answer and correction;

According to the reviewer`s comment, we added the following sentence in line 63-67 as “PCT is useful for the diagnosis of sepsis. In the presence of bacterial infection, PCT is rapidly produced by the C cells of the thyroid gland as well as several other cell type. PCT production is stimulated by two mechanisms, directly by bacterial endotoxins and lipopolysaccharides and indirectly by inflammatory mediators such as tumor necrosis factor-alpha, interleukin-6, interleukin-1 [9, 10]” 

2- Fever is responsible to increase serum level of PCT or PCT is responsible for fever? 

Answer and correction;

Thank you for your careful review. PCT level increases as a result of endotoxins and/or cytokines, and fever happens as a systemic response to any kind of sources like infection or inflammation. Both fever and increase of PCT level might be the results of infection. We described that in introduction as “The key manifestation of infection is fever, although various noninfectious febrile episodes can also develop frequently. In HSCT patients, it is more complex to distinguish between infectious condition and aseptic causes of febrile events due to transplantation-related complications such as graft-versus-host disease, engraftment syndrome, thrombotic microangiopathy, and relapse of underlying diseases” line 51-56.

3- However, usually patients with malignant hematopoiesis disorder have fever and immune system of this kind of patient is not effective, therefore infection is as usual. May be the cause of increase serum level of PCT, were not be related to malignant hematopoiesis disorder.

The Authors find that the PCT serum level increase by malignant hematopoiesis disorder or transplantation, here we should consider to some question:

a- In other malignancies what is happened for PCT? It has been necessary to have data from other malignancy and discusses about that.

Answer and correction;

In solid cancer, similar results were observed. According to the reviewer’s comments, we added that as “Previous reports have shown that PCT has high specificity for bacterial infections, and PCT has been introduced into clinical use [21.22]. Kim et al., reported that PCT showed better diagnostic value than CRP in febrile neutropenic patients with solid cancer [19].” in discussion, lines 282-285.

b- Is it possible PCT would be prognosis factor or treatment evaluated factor?

Answer and correction;

Although the present study did not evaluate the prognostic role of PCT, previous studies revealed the prognostic values of PCT in patients with hematologic malignancies. We described in introduction as “It could also predict patient’s prognosis after HSCT [1, 10].” “ in lines 71-72. 

The discussion should be more and improve.

Answer and correction;

According to the reviewer’s comments, we added description as “As previous studies reported that PCT was useful in monitoring the response to the infection and to the treatment [1, 10], prognostic role of PCT needs to be further explored in a large, well-defined cohort study.” in lines 343-345.

Reviewer #2: The authors present an analysis of a large patient cohort with haematological diseases focusing on infection monitoring with laboratory test such as CRP and PCT in addition to microbiological examinations. However, the presentation of the data in its current form is poor, lacks clarity, important details and interpretation.

Major:

1. It should be elaborated if there are overlaps between patients with hematology diagnosis and those that were treated by HSCT. What were the typical diagnoses for HSCT?

Answer and correction;

Thank you for your careful review. According to the reviewer’s comments, we added the that as ”The primary diagnoses of 325 HSCT patients were AML (n=135), ALL (n=75), lymphoma (n=44), multiple myeloma (n=44) and myelodysplastic syndrome (n=17).” In lines 138-139. 

2. More details should be provided about viral infections and non-infectious etiology: lines 145 and 146.

Answer and correction;

According to the reviewer’s comments, we added detailed diagnoses as “Non-bacterial infections were identified in 515 (83.9%) febrile episodes caused by fungal infection (n = 29, 4.7%), viral infection (n = 28, 4.6%) or other noninfectious etiology (n = 458, 74.6%). The virus infections belong to the cytomegalovirus (n = 19), Human herpes virus (n = 5) and influenza virus (n = 4). Noninfectious febrile etiologies included pulmonary complication (n = 113), included local infection (n = 35), engraft syndrome (n = 13), enterocolitis (n = 11), graft versus host disease (n = 8), relapse (n = 8) and miscellaneous or unknown source (n = 270).” in lines 141-147. 

3. Statements of lines 163 and 164 do not correspond with the numbers of Table 2.

Answer and correction;

According to the reviewer’s comments, we changed the sentence into “Of 99 febrile episodes with bacteremia, 65 episodes were from HSCT patients (20.0%, 65/325) and 34 episodes were from non-HSCT patients (11.8%, 34/289) (P = 0.006).” in lines 156-158.

4. Figure legends in general must be substantially extended with additional details e.g. case numbers in the respective subgroups, rational and method of subgrouping, statistics employed etc. I do not clearly understand the difference between Figs 1 and 2, similarly the meaning of colour is also missing.

Answer and correction;

According to the reviewer’s comments, we changed figure legends as

” Fig 1. Comparison of PCT (left) and CRP (right) levels between bacteremia and non-bacteremia. Ninety-nine systemic bacterial infection episodes (colored box) showed higher PCT and CRP levels than nonbacterial events [median (95% CI) (PCT: 0.49 (0.26 - 0.93) ng/mL vs. 0.20 (0.18 – 0.22) ng/mL, P < 0.001; CRP: 76.6 (50.5 – 92.8) mg/L vs. 58.0 (51.1 – 66.5) mg/L, P = 0.036)] by Mann-Whitney U test.

Fig 2. Comparison of PCT (left) and CRP (right) levels between neutropenic and non-neutropenic patients. PCT levels discriminated bacteremia (gray box) from non-bacteremia (white box) in both neutropenia (-) patients [median (95% CI), 0.97 (0.27 – 5.61) ng/mL vs. 0.20 (0.17 – 0.24) ng/mL, P < 0.001] and neutropenia (+) patients [0.44 (0.19 – 0.78) ng/mL vs. 0.20 (0.18 – 0.22) ng/mL, P = 0.001]. CRP levels discriminated bacteremia (gray box) from non-bacteremia (white box) in neutropenia (-) patients [146.1 (78.9 – 240.2) mg/L vs. 52.2 (46.4 – 62.0) mg/L, P < 0.001)], but CRP levels were not different between bacteremia and non-bacteremia in neutropenia (+) patients [59.7 (41.7 -84.5) mg/L vs. 69.5 (54.6 – 79.0) mg/L, P = 0.994] by Mann-Whitney U test.” in lines 183-197. 

“Fig 3. Comparison of PCT (left) and CRP (right) levels between HSCT and non-HSCT. PCT levels discriminated bacteremia (gray box) from non-bacteremia (white box) in both non-HSCT patient [median (95% CI) 0.91 (0.27 – 5.51) ng/mL vs. 0.22 (0.18 – 0.28) ng/mL, P < 0.001] and HSCT patients [0.30 (0.18 – 0.58) ng/mL vs. 0.19 (0.16 – 0.22) ng/mL, P < 0.001]. CRP levels discriminated bacteremia (gray box) from non-bacteremia (white box) in non-HSCT patient [125.7 (91.3 – 159.8) mg/L vs. 67.4 (55.7 – 81.9) mg/L, P = 0.002] but CRP levels were not different between bacteremia and non-bacteremia in HSCT patients [51.0 (39.5 – 78.4) mg/L vs. 49.3 (39.9 – 61.3) mg/L, P = 0.414] by Mann-Whitney U test.” in lines214-222.

“Fig 4. ROC curves of PCT and CRP for detecting bacteremia in total (a), non-neutropenic (b), and neutropenic (c) patients. In a total of 614 febrile episodes (a), the AUC of PCT was 0.651 (95% CI: 0.612 – 0.689) while that of CRP was 0.566 (95% CI: 0.526 -0.606). In 341 non-neutropenic episodes (b), PCT and CRP showed AUC of 0.757 (95% CI: 0.708-0.801) and 0.763 (95% CI: 0.714 – 0.807), respectively. In 273 febrile neutropenia (c), PCT discriminated bacteremia from non-bacterial infection (AUC: 0.624, 95% CI: 0.564 - 0.682) whereas CRP could not detect bacteremia (AUC: 0.500, 95% CI: 0.439 – 0.561, P > 0.05).” in lines 241-248.

“Fig 5. Comparison of PCT (left) and CRP (right) levels among different etiologies of fever. PCT levels were significantly higher in GNB infectious episodes (gray box) than those in febrile episodes caused by fungal infection [median (95% CI), 0.58 (95% CI: 0.26-1.61) vs. 0.22 (0.16 – 0.38), P = 0.047] or other etiology [0.58 (95% CI: 0.26-1.61) vs. 0.19 (0.17 – 0.22), P < 0.001]. CRP levels were not significantly different among the four groups (P > 0.05).” in lines 267-272.

5. In addition to the above deficiencies, in the context of Fig 5 (lines 239 through 248), respective case numbers are missing. There is an apparent discrepancy between CRP differences in earlier comparisons and the apparent similarities in these comparisons.

Answer and correction;

According to the reviewer’s comments, case numbers are added in the context and figure as “PCT and CRP levels in four different febrile etiologies (38 GPC, 61 GNB, 29 fungus and 486 others etiologies) are shown in Figure 5.” in lines 255-256 and figure 5.

We agree with reviewer`s opinion. In Figure 1, CRP levels were higher in bacteremia (+) group compared to the bacteremia (-) group (P =0.036), However, when we compared the CRP levels among four groups [GNP(+), GNB(+), fungus (+), others(+)], there was no significant difference. In addition, there were no significant differences in PCT or CRP level between Gram-positive and Gram-negative infectious episodes (P > 0.05). We clarified that as “Although bacteremia (+) patients had higher CRP levels than bacteremia (-) patients, CRP levels were not significantly different among the four groups (P > 0.05). There were no significant differences in PCT or CRP level between Gram-positive and Gram-negative infectious episodes (P > 0.05).” in lines 259-261.

6. The abstract should be completely rewritten. For me the line of study is unclear. In my opinion a single observation is arbitrarily emphasized as conclusion.

Answer and correction;

Thanks for your comments. We tried our best to improve the manuscript and made some correction in abstract according to the reviewer’s comments and emphasized in conclusion as “Conclusions: In this single-center observational study, PCT was more valuable than CRP for discriminating between bacteremia and non-bacteremia independent of neutropenia or HSCT.” 

Minor: potential points of grammatical improvements

1. Table 1 should be omitted, it is redundant relative to Table 2

Answer and correction;

According to the reviewer’s comments, table 1 in submitted manuscript was deleted.

2. In Table 2 ANC value (median) <0.01 is unlikely to be correct for Bacteremia pos pts

Answer and correction;

According to the reviewer’s comments, we corrected that.

3. Table 2 abbreviations should be in alphabetical order

Answer and correction;

According to the reviewer’s comments, we corrected them in alphabetical order.

4. English proficiency should be further improved. Even myself, a non-English native, found several incorrectly worded phrases.

Answer and correction;

We have undergone English edit procedure. 

I hope the revised manuscript will better meet the requirements of the ‘Plos one’ for publication. I thank you again for the constructive review by the referee.

Eun-Jee Oh

---

## [Decision Letter · Decision Letter 1]

13 Nov 2019

Serum procalcitonin as an independent diagnostic markers of bacteremia in febrile patients with hematologic malignancies

PONE-D-19-16377R1

Dear Dr. Oh,

We are pleased to inform you that your manuscript has been judged scientifically suitable for publication and will be formally accepted for publication once it complies with all outstanding technical requirements.

With kind regards,

Senthilnathan Palaniyandi, Ph.D

Academic Editor

PLOS ONE

Additional Editor Comments (optional):

Reviewers' comments:

Reviewer's Responses to Questions

**Comments to the Author**

1. If the authors have adequately addressed your comments raised in a previous round of review and you feel that this manuscript is now acceptable for publication, you may indicate that here to bypass the “Comments to the Author” section, enter your conflict of interest statement in the “Confidential to Editor” section, and submit your "Accept" recommendation.

Reviewer #2: All comments have been addressed

2. Is the manuscript technically sound, and do the data support the conclusions?

Reviewer #2: Yes

3. Has the statistical analysis been performed appropriately and rigorously? 

Reviewer #2: Yes

4. Have the authors made all data underlying the findings in their manuscript fully available?

Reviewer #2: Yes

5. Is the manuscript presented in an intelligible fashion and written in standard English?

Reviewer #2: Yes

6. Review Comments to the Author

Reviewer #2: The questions raised have been adequately answered and appropriate modifications have bben made to the text.

7. PLOS authors have the option to publish the peer review history of their article (what does this mean?). If published, this will include your full peer review and any attached files.

Reviewer #2: No

---

## [Editor Report · Acceptance letter]

20 Nov 2019

PONE-D-19-16377R1 

Serum procalcitonin as an independent diagnostic markers of bacteremia in febrile patients with hematologic malignancies 

Dear Dr. Oh:

I am pleased to inform you that your manuscript has been deemed suitable for publication in PLOS ONE. Congratulations! Your manuscript is now with our production department. 

With kind regards,

on behalf of

Dr. Senthilnathan Palaniyandi 

Academic Editor

PLOS ONE